# Liquid Biopsy and Challenge of Assay Heterogeneity for Minimal Residual Disease Assessment in Colon Cancer Treatment

**DOI:** 10.3390/genes16010071

**Published:** 2025-01-09

**Authors:** Giovanni Crisafulli

**Affiliations:** IFOM—The AIRC Institute of Molecular Oncology, 20139 Milano, Italy; giovanni.crisafulli@ifom.eu

**Keywords:** colorectal cancer, liquid biopsy, minimal residual disease, precision oncology, ctDNA

## Abstract

This review provides a comprehensive overview of the evolving role of minimal residual disease (MRD) for patients with Colon Cancer (CC). Currently, the standard of care for patients with non-metastatic CC is adjuvant chemotherapy (ACT) for all patients with stage III and high-risk stage II CC following surgical intervention. Despite a 5–20% improvement in long-term survival outcomes, this approach also results in a significant proportion of patients receiving ACT without any therapeutic benefit and being unnecessarily exposed to the risks of secondary side effects. This underscores an unmet clinical need for more precise stratification to distinguish patients who necessitate ACT from those who can be treated with surgery alone. By employing liquid biopsy, it is possible to discern MRD enabling the categorization of patients as MRD-positive or MRD-negative, potentially revolutionizing the management of ACT. This review aimed to examine the heterogeneity of methodologies currently available for MRD detection, encompassing the state-of-the-art technologies, their respective advantages, limitations, and the technological challenges and multi-omic approaches that can be utilized to enhance assay performance. Furthermore, a discussion was held regarding the clinical trials that employ an MRD assay focusing on the heterogeneity of the assays used. These differences in methodology, target selection, and performance risk producing inconsistent results that may not solely reflect biological/clinical differences but may be the consequence of the preferential use of particular products in studies conducted in different countries. Standardization and harmonization of MRD assays will be crucial to ensure the liquid revolution delivers reliable and clinically actionable outcomes for patients.

## 1. Introduction on the Clinical Unmet

The clinical management of patients with non-metastatic colorectal cancer (CRC) necessitates a multidisciplinary approach that integrates surgery, systemic oncological therapies, and, in select cases, radiotherapy [1,2]. This approach is dependent on the tumor stage, location, and molecular characteristics [2,3,4]. Surgery represents the primary treatment for non-metastatic Colon Cancer (CC) and has the potential to be curative, with success rates varying according to the stage of the disease. For patients with high-risk stage II or stage III tumors, surgical resection is frequently followed by adjuvant chemotherapy (ACT), typically administered over a period of 3 to 6 months, with the objective of minimizing the risk of local or metastatic recurrence [1]. Despite the implementation of ACT, recurrence rates remain significant, with approximately 15–20% of patients with stage II CRC and 30% of those with stage III CRC experiencing relapse [1,5]. Surgery is curative in approximately 80% of stage II cases and 50% of stage III ones [5,6]. These statistics highlight the pressing need for more efficacious therapeutic strategies to prevent recurrence through the implementation of precision medicine approaches [5]. Figure 1 summarizes and schematizes the concepts described above, based on the current literature, and the differences between stage II–III CC [5,7,8,9,10].

Furthermore, as a substantial proportion of patients may achieve definitive cure through surgical intervention alone, the risk of overtreatment must be carefully considered and critically evaluated. For these individuals, adjuvant chemotherapy offers no clinical benefit, while potentially subjecting them to toxic side effects and complications that impair quality of life and increase healthcare costs [7] (Figure 1).

A novel objective in the clinical management of resectable CC is the identification and development of biomarkers capable of detecting minimal residual disease (MRD) following surgery [11,12].

A question thus arises: What is MRD? MRD is defined as the persistence of neoplastic cells that are microscopically undetectable yet still present in the circulation after an apparently curative treatment [13]. These cells have the potential to drive disease recurrence. This microscopic nature underscores a significant limitation of conventional diagnostic techniques, such as computed tomography (CT) and other radiological imaging modalities, which lack the sensitivity to detect residual microscopic disease. These methods are unable to identify the presence of residual tumor cells post-surgery that fall below the detection thresholds of standard technologies [14,15].

The presence of MRD is recognized as the biological foundation of metastatic relapse, and its early detection is increasingly seen as pivotal for guiding postoperative therapeutic decisions [16]. Parallelly, the modulation of the intensity of adjuvant therapies, with the dual objective of enhancing patient outcomes and reducing the unnecessary burden of treatment is yet under study (December 2024) [17,18,19]. By addressing MRD, clinicians can potentially pursue one of the key objectives of personalized medicine, namely, providing the optimal treatment at the appropriate time to the most suitable patient: evaluating whether ACT can be spared when unnecessary and provided or even intensified when needed to maximize outcomes [20,21,22,23].

## 2. Technology for the Detection of Minimal Residual Disease

Liquid biopsy is a minimally invasive method that extracts genetic and multi-omic information from a simple blood draw [24,25]. It is a cost-effective method of medical analysis that is less invasive than traditional tissue analysis methods having the capacity to overcome the limitations of tissue-based methodologies, such as the effects of tumor heterogeneity and sampling bias [26,27]. However, like all technologies, liquid biopsy has its limitations. It cannot provide information about tissue architecture and some cellular characteristics, and it may suffer from variability in ctDNA shedding [25,28,29,30].

Liquid biopsy techniques represent a promising innovation for the detection of MRD in CC. These approaches employ a range of omics methodologies, with genomics and methylomics being the most prevalent [31,32]. Genetic approaches are frequently employed in isolation or in conjunction with other omics techniques, such as epigenetics, due to the relatively high release of circulating tumor DNA (ctDNA) observed in CC [17,33]. Indeed, CC is among the cancer types that exhibit a substantial release of ctDNA into the bloodstream, in stark contrast to malignancies like gliomas or renal cell carcinoma, which exhibit low ctDNA levels in blood [29,34,35]. It is also known that tumor size and proliferative capacity seem to be drivers of ctDNA release in CC [36]. Prostate cancer is an exemplar of a neoplasm that was formerly classified as a low releaser [29]. However, the recent literature suggests a reclassification of this neoplasm as a high releaser [37].

The scientific basis of ctDNA-based MRD assays is the analysis of ctDNA, which comprises fragments of tumor-derived DNA shed into the bloodstream by residual tumor cells [14]. This method is minimally invasive, relying solely on a simple peripheral blood draw, and offers a sensitive and specific means of evaluating disease persistence at the molecular level [12,20,38,39]. In contrast to conventional imaging techniques such as CT or other radiological methods, ctDNA analysis enables the detection of MRD at resolutions below the limits of these traditional approaches [14,15,40]. Furthermore, ctDNA functions as a dynamic biomarker for the real-time monitoring of therapeutic response and the assessment of recurrence risk [14,15,41].

A crucial element of ctDNA-based MRD detection is the timing of blood sample collection in relation to surgical procedures. The current literature suggests that blood samples for MRD analysis should be collected no earlier than two weeks after surgery, with an optimal window of five weeks, even if 6/8/12 weeks could be reached [42] (Figure 2). Earlier sampling may result in the detection of an excess of DNA released during the surgical resection process, which could dilute the ctDNA signal with an excessive amount of DNA derived from normal cells. Additionally, ACT has been observed to reduce ctDNA levels, necessitating the collection of samples prior to initiating systemic therapies to ensure accuracy and allow treatment decisions [21].

One of the major challenges in liquid biopsy for MRD is the inherently low tumor content in the blood, defined as the ratio of ctDNA to cell-free DNA (cfDNA) [25,43]. The ratio of ctDNA to cfDNA can vary considerably, from 0.05% to 90% [25,44,45,46]. However, it is often biased in favour of cfDNA, which presents a significant challenge for assays designed to detect rare ctDNA fragments. This underscores the importance of the limit of detection (LOD) of these assays, as they must identify small amounts of ctDNA among millions of cfDNA molecules derived from normal cells [25,47].

To address this challenge, MRD assays typically employ two primary strategies. The first approach entails the utilization of “plasma-only” assessments, which concentrate on a predefined set of genes frequently mutated in CC (or, broadly, CRC), as evidenced by the scientific literature [38,48,49]. By focusing on a specific set of genomic regions, these tests can sequence these regions with a high depth, thereby improving sensitivity and specificity, often leveraging systems such as DNA barcoding to enhance detection capabilities [50,51,52,53] (Figure 2).

An alternative approach is the “tumor-informed” test, which adopts a personalized methodology. These assays initially examine the tumor tissue of a given patient to identify somatic mutations specific to the individual’s cancer. The identified mutations are then selectively searched for in the cfDNA from the patient’s blood sample, enabling precise MRD detection (Figure 2).

Both approaches address the tumor content/LOD issue by tailoring the analysis to maximize the likelihood of ctDNA detection while minimizing interference from cfDNA. However, they also reflect the broader technological and clinical challenges in implementing liquid biopsy-based MRD detection, emphasizing the need for further optimization and validation to fully integrate these assays into clinical practice [34].

A disadvantage common to both plasma-only and tumor-informed tests, however, arises from the reliance on genetic profiling as the foundation pillar for classification. Indeed, metastases located in specific organs, such as the lungs or peritoneum, are known to release low levels of ctDNA [54,55]. Thus, the biological and anatomical characteristics of a tumor recurrence/persistence can lead to false-negative results, as the low ctDNA output may fall below the detection threshold of even the most advanced MRD assays [54]. This limitation highlights the importance of considering tumor biology and metastatic site characteristics in the interpretation of MRD test results and underscores the need for complementary approaches to improve the robustness of liquid biopsy strategies.

### 2.1. Features of Plasma-Only or Tumor-Agnostic Assay

Plasma-only assays are available for the detection of MRD using blood samples. These assays analyse cfDNA without the necessity of prior tumor tissue sequencing. The most prevalent product in the market is the Guardant Reveal™ assay, a liquid biopsy product designed to detect MRD and monitor recurrence in locally advanced CRC patients (Table 1). This test attains high sensitivity by concurrently analyzing genomic alterations and methylation patterns, thereby yielding results from a simple blood draw without the necessity for tissue samples [33]. It is important to highlight that the new version of the Reveal assay is based only on 20,000 epigenomic regions and does not encompass a genetic panel anymore (Table 1).

In addition to methylation patterns, plasma-only assays can employ methodologies based on fragmentomics. It is known that ctDNA fragments are typically shorter than cfDNA ones originating from healthy cells in the blood as in other fluid such as urine [56,57]. This distinction in the fragment dimensions and associated physicochemical characteristics can be utilized to enhance the detection of MRD. The application of fragmentomics enables these assays and could improve their capacity to predict MRD status with greater precision [58,59]. Among commercially available plasma-only assays, the AVENIO ctDNA Surveillance Kit V2™ stands out for its advanced design and the possibility to use blood or urine samples. It utilizes the iDES system, which combines molecular barcoding with in silico error suppression techniques to ensure high accuracy [53]. The assay targets 197 genes, focusing on 471 regions frequently mutated in diseases, including genes listed in the National Comprehensive Cancer Network (NCCN) guidelines [3,4]. AVENIO is commercialized by Roche for targeted detection of single nucleotide variants (SNVs), insertions and deletions (indels), fusions, and copy number alterations (CNAs) [60]. A similar test that may work with blood and urine samples is the PredicineALERT™ MRD assay, but in this case the specific genomic target regions are not reported, even though the platform PredicineATLAS™ (2022) was reported to be based on a 600-gene hybrid capture-based NGS assay by Chen and colleagues [33] (Table 1 and Figure 2).

Beyond fragmentomics, another type of analysis that could increase the MRD detection is the incorporation of mutational signature information into MRD assays. This could provide an additional degree of accuracy. This analysis is based on the concept that tumor mutational signatures exhibit distinctive differences compared to healthy profiles [61,62]. These differences could be exploited to differentiate between healthy and diseased individuals in the future.

Like mutational signature analysis, all approaches that can stratify healthy donors from patients have the potential to markedly enhance the accuracy of relapse predictions, making it a promising avenue for further research and clinical application.

A specific constraint associated with the plasma-only assay pertains to the dimensions of the genetic target under scrutiny. It stands to reason that commercial products exploring broad genetic targets have a higher likelihood of detecting ctDNA compared to plasma-only products based on smaller genetic targets. These smaller genetic targets may fail to capture the specific genetic alterations of each type of tumor (Table 1).

### 2.2. Features of Tumor-Informed Assays

Tumor-informed MRD assays offer a personalized diagnostic approach for detecting ctDNA in blood samples by first identifying tumor-specific mutations through genomic sequencing of tumor tissue. These assays are characterized by a two-step methodology: an initial exploratory sequencing phase to identify somatic mutations in the tumor (spanning the genome, exome, or large targeted gene panels) followed by a targeted analysis of ctDNA in plasma.

A key differentiating factor among these assays is the genomic target analysed in the tumor tissue and the subsequent number of tumor-specific variants monitored in blood. When not protected by proprietary restrictions, it is known that the genomic target may vary from whole-genome sequencing (WGS), whole-exome sequencing (WES), or large targeted panels (Table 1). The number of specific variants tracked in plasma ranges from thousands to a much smaller, carefully curated subset, depending on the assay. For example, MRDetect™ and PlasmaDetect™ focus in whole-genome somatic variants, RaDaR™ focuses on 48 tumor-specific variants, and Signatera [63,64] uniquely monitors 16 clonal tumor-specific variants [33,39,65]. In addition, not only Next Generation Sequencing (NGS), but also droplet digital polymerase chain reaction (ddPCR) methods may be used in a similar manner for the detection of genetic alterations in the blood [17,33].

MRDetect, developed by the Landau lab (led by Dr. Asaf Zviran), is an ultra-sensitive, machine learning–powered, tumor-informed DNA-sequencing strategy designed for MRD detection [66]. By analyzing WGS data from tumor samples, MRDetect searches for cumulative patterns of mutations and CNAs in ctDNA [67]. In detail, the algorithm optimizes sensitivity by balancing breadth (number of mutations sequenced) and depth (sequencing coverage of depth) and introduces a read-centric framework to distinguish true variants from sequencing artifacts [68,69]. According to the authors, MRDetect may predict tumor fraction (TF) in cfDNA based on the number of detected sites, mutation load, and sequencing depth. Independent classifiers for SNVs and CNAs are integrated into a single detection score, improving detection power even in ultra-low variant allele frequency (VAF) cases. This approach should overcome ctDNA sampling limitations and delivers high accuracy for MRD detection [66]. PlasmaDetect is another product based on a similar approach (Table 1).

In conclusion, it is relevant to report that proprietary assays may use unknown targets, further diversifying their approach such as Haystack MRD or RADAR by Neogenomics (Table 1 and Figure 2).

Even if these tumor-informed assays are very heterogeneous, they represent a significant advancement in personalized oncology, allowing clinicians to adapt treatment strategies to the molecular profile of each patient’s cancer. By leveraging tumor-specific information, these tools improve the precision of MRD detection and offer the potential to optimize therapeutic interventions [42].

However, a key disadvantage of tumor-informed assays compared to plasma-only tests lies in their experimental design and clinical applicability. Since these assays classify MRD based on genetic variants identified in the tumor, they are subject to certain limitations. The first challenge arises with tumors that exhibit high heterogeneity [26]. In such cases, sequencing may reveal mutations that represent “clonal illusions”, meaning that they are present at clonal level in the sequenced tissue but do not accurately reflect the entire tumor’s genetic makeup [70]. This could lead to misclassifications and a higher likelihood of false negatives in MRD detection, as the assay would be based on a subset of mutations that may not be universally present in the tumor (Table 1).

A second issue arises from the potential evolution of the tumor over time, particularly when there is a significant delay between tumor tissue analysis and the MRD test using blood. As the tumor evolves, it may lose certain genetic characteristics, including those present at the clonal level in the initial tumor sample [14]. This evolution could lead to the disappearance of certain mutations that were originally used to define the tumor signature, affecting the accuracy of the MRD test in detecting disease recurrence (Table 1).

The last issue in tumor-informed MRD assays arises when tumor tissue is unavailable, which is increasingly common in cases where a neoadjuvant approach is pursued. For instance, in MSI-high colorectal cancers, patients often receive adjuvant immunotherapy, and in rectal cancer, neoadjuvant chemoradiotherapy is a standard of care. These treatments can lead to a complete pathological response, where no residual tumor tissue remains post-surgery. In such cases, the surgical specimen, typically used to identify tumor-specific mutations for MRD assays, cannot serve this purpose. The alternative may be the utilization of the diagnostic biopsy obtained at the time of diagnosis. However, biopsies often provide limited material, which may not be sufficient for comprehensive genomic profiling. This limitation makes tumor-informed approaches unusable in these scenarios (Table 1).

Thus, while tumor-informed assays provide personalized insights, their accuracy can be compromised by tumor heterogeneity and evolutionary changes over time and in specific cases tumor-informed approaches may be unusable when tumor tissue is unavailable.

## 3. Is the Liquid Biopsy Revolution Ready? Insights from Clinical Trials

The clinical utility of liquid biopsy in MRD detection is currently being rigorously evaluated in numerous ongoing and recently concluded clinical trials. The objective of these studies is, first, to establish the clinical utility of MRD stratification in guiding adjuvant treatment choices, and, in particular, to determine whether liquid biopsy can guide the intensification or deintensification of adjuvant therapy following surgery in CC patients [16,71].

The initial breakthrough evidence was presented in 2022 by Tie et al. [16], who demonstrated in the randomized phase II DYNAMIC trial the non-inferiority in terms of recurrence-free survival of a liquid biopsy-driven approach to inform adjuvant treatment decisions in stage II CC patients. Building on this, the phase II PEGASUS trial (NCT04259944), presented by Lonardi et al. at the ESMO Congress 2023 [72], extended this evaluation to high-risk stage II (pT4N0) and stage III CC, highlighting that liquid biopsy may be used to guide the post-surgical clinical management of CC patients by reducing unnecessary toxicity and by improving the response to standard chemotherapy. The findings of this study are grounded in evidence showing that patients randomized to the ctDNA-guided arm achieved similar survival outcomes while receiving approximately half the therapy compared to the physician-choice arm. This highlights the potential of liquid biopsy to safely reduce chemotherapy exposure without compromising patient prognosis, offering a significant step forward in personalized cancer care [72].

In the next years, ongoing and newly designed randomized trials will define the clinical utility of liquid biopsy for MRD in CC. Up to December 2024, a large number of clinical trials were investigating the potential of ctDNA to inform the stratification of MRD after surgery. In this context, notable examples include the randomized DYNAMIC III (ACTRN12617001566325), CTAC (NCT05529615), and CIRCULATE US (NCT05174169) trials, which are investigating the potential of MRD results to inform decisions regarding the escalation or de-escalation of adjuvant therapy. For example, these studies assess whether intensifying therapy in patients with ctDNA-positive results or de-escalating treatment in patients with ctDNA-negative results might improve outcomes and spare toxicity [17,39].

Moreover, other clinical trials are specifically focused on treatment de-escalation: The UK TRACC (NCT04050345) and CIRCULATE-Japan (UMIN000039205) trials aim to demonstrate the non-inferiority to the standard of care in terms of three-year disease-free survival (DFS) [73,74]. Moreover, the CIRCULATE-Japan trial will assess the efficacy of a combination of capecitabine and oxaliplatin (CAPOX) regimen in MRD-negative patients, with the objective of reducing the treatment burden without compromising efficacy [75,76].

Concurrently, other studies will ascertain the efficacy of therapy intensification. Indeed, the GALAXY study provided compelling evidence that ctDNA-positive patients are at a markedly elevated risk of recurrence compared to those who test ctDNA-negative [77]. This has prompted the initiation of clinical trials, such as ERASE CRC trial (NCT05062889) and AFFORD (NCT05427669), with the objective of evaluating the potential benefits of intensified therapeutic regimens. Also, the previously cited PEGASUS trial has studied the feasibility of upscaling therapy in ctDNA-positive patients, wherein treatment regimens such as folinic acid, fluorouracil and irinotecan combination (FOLFIRI) or CAPOX were tailored based on ctDNA results [72]. Similarly, the ERASE CRC compares the efficacy of the folinic acid, fluorouracil and oxaliplatin combination (FOLFOX)/CAPOX with that of the oxaliplatin, irinotecan and fluorouracil one (FOLFOXIRI) in patients with MRD-positive disease, with ctDNA clearance designated as the primary endpoint [17].

Additionally, novel strategies, including targeted treatments, are currently being explored. In the ERASE CRC study, a subgroup of patients with a HER2+/RAS wild-type ctDNA-positive status receives FOLFOX in combination with trastuzumab and tucatinib, as another specific example of therapy intensification in the targeted therapy context. Other trials, such as CIRCULATE Spain (EudraCT Number: 2021–000507-2), investigate dual-arm approaches to compare intensified regimens (e.g., FOLFOXIRI vs. CAPOX) in MRD-positive patients [17].

These examples represent a mere sampling of the ongoing research aimed at refining treatment strategies based on MRD detection. Additional trials, including PRODIGE 70 CIRCULATE (NCT00002019-000935–15), IMPROVE-IT (NCT03748680), MEDOCC-Create (NL6281/NTR6455), SAGITTARIUS (NCT06490536), and CIRCULATE AIO-KRK-0217, (NCT04089631) provide further evidence of the global effort to optimize CRC management through ctDNA-guided approaches [17,78,79].

Various MRD assays have been performed in presented clinical trials, showing that the post-surgery proportion of ctDNA-positive patients with stage I–III CRC ranges from 6%—stage II: plasma-only Guardant Health—to 29%—stage I–IV: tumor-informed custom (Table 2) [80]. It is important to note that both the liquid biopsy assays, stages, and time points for evaluation of MRD were highly heterogeneous in existing studies (Table 2) [48,81,82,83,84]. Among the most commonly used plasma-only tests, Guardant Health emerges as a leading choice, with its various versions being utilized in 60% (three out of five) studies [72,85]. In contrast, among the tumor-informed tests, Signatera stands out as the preferred option in 41% (five out of twelve) cases [77,86,87]. Among custom assays, the Safe-SeqS assay is noteworthy, having been employed for exploratory purposes to identify mutations to be tracked in plasma (custom tumor-informed assay) in the three studies led by Tie and colleagues (Table 2) [16,88,89].

A thorough review of the clinical trials also reveals that there was a preference for certain approaches based on the geographic location of the studies. For example, French studies exhibited a tendency to favour custom plasma-only approaches, as illustrated in the works by Taieb and Benhaim and colleagues [82,83]. Conversely, Danish studies employed tumor-informed approaches, as evidenced by Scholer, Reinert, Henriksen and colleagues [80,86,87], with Signatera being the assay of choice in two out of three cases [86,87]. Furthermore, the three studies conducted by Tie and colleagues (2016, 2019, 2022) also favour a tumor-informed approach, with their use of the Safe-SeqS assay [16,88,89] (Table 2).

Despite the limited number of trials included in the analysis (typically two or three per country), the findings suggest that there are research group preferences for specific approaches and assays that may be influenced by their individual experiences and areas of expertise. The Australian case, exemplified by the consistent use of custom tumor-informed assays in Tie’s studies, is particularly emblematic of this trend. In Table 2, I reported the percentages of MRD positivity from published clinical trials as well as from clinical trials whose results have been officially presented at international meetings but have not yet been published in high-impact journals [72,90,91,92,93].

## 4. Discussion

The aim of this review was to provide a comprehensive overview of the evolving role of MRD for patients with CC, addressing the critical issue of assay heterogeneity in MRD assessment for CC treatment. Naturally, the primary limitation of this review was in the inability to access detailed technical specifications of the individual commercial MRD products. Due to intellectual property protection, companies often do not disclose these details. As a result, while I aimed to provide a precise evaluation of the differences between various commercial products, it was not always possible to explore the specifics of each product. Consequently, this review focused on the two overarching approaches that form the basis of these assays: tumor-informed and plasma-only.

Since the pivotal study by Tie et al. in 2022 [5], the clinical validity/utility of liquid biopsy in stratifying patients after surgery based on the presence or absence of MRD has been a central matter of clinical research in oncology. Currently, there are at least 25 ongoing or recently concluded clinical trials investigating the utility of liquid biopsy to guide the escalation or de-escalation of adjuvant therapy based on MRD results [17]. The sheer number of clinical trials underscores the clinical relevance of this approach and suggests that a “liquid revolution” in this context is on the horizon. The ultimate goal of these efforts is highly noble: to avoid overtreatment and the associated side effects of unnecessary therapies for MRD-negative patients, while improving therapy outcomes for MRD-positive patients at high risk of relapse (i.e., through intensification or therapeutic switch in micrometastatic resistant cases).

Despite the scientific progress based on the clinical trial results, several critical challenges remain unresolved. There is currently no standardized assay to determine whether a patient is micrometastatic (MRD+) or not (MRD−) after surgery. The available MRD tests can be broadly classified into tumor-agnostic (or plasma-only) and tumor-informed assays. Plasma-only tests analyse patient plasma without relying on tumor tissue, while tumor-informed tests utilize tumor tissue to identify mutations that are specific of a given tumor that are then sought in the same patient’s blood. Such heterogeneity and lack of standardization pose significant obstacles to consistent results and clinical adoption.

At first glance, plasma-only tests may appear more standardized due to their tumor-agnostic nature, but this is not always the case. Some plasma-only tests have evolved over time, shifting from the combination of genomic and epigenomic analyses to exclusive epigenetic profiling, such as the Guardant 2024 update example. Similarly, tumor-informed assays exhibit significant variability in design, relying on tumor-specific mutations identified through different methodologies, ranging from WGS to targeted hotspot panels. The techniques used to monitor these mutations in blood also vary widely, including ddPCR, amplicon-based NGS, and capture-based NGS panels.

Biological biases further complicate tumor-informed assays due to the inherent heterogeneity of CRC. Tumor heterogeneity can lead to the phenomenon of “clonal illusion”, where exploratory sequencing of the tumor tissue fails to identify truly representative clonal variants, resulting in false negatives. Moreover, tumor evolution poses another challenge, as these assays confines their analysis at the mutations identified in the tumor tissue, rendering them incapable of detecting new mutations acquired over time. This limitation negates one of the primary advantages of liquid biopsy: the ability to capture tumor heterogeneity and dynamics in real time. Additionally, we must not forget that tumor tissue may sometimes be unavailable, making it impossible to use tumor-informed assays.

Another relevant point to highlight is that CRC also exhibits variability in ctDNA release based on metastatic sites, adding another layer of complexity. Studies indicate that metastases in the lungs and peritoneum release less ctDNA than those in the liver, creating additional difficulties for MRD tests aiming to classify patients as MRD+ or MRD−. Moreover, the timing of post-surgical blood draws introduces further variability, with different trials collecting samples between 2 and 6–12 weeks after surgery. Also in this context, to date, no studies have systematically explored the dynamics of ctDNA release during this critical postoperative window, leaving a significant gap in knowledge.

This combination of pivotal clinical interest and lack of standardization in MRD testing raises the risk of generating inconsistent results, driven not by biological or clinical differences but by technical variability. For example, preliminary data indicate that the proportion of ctDNA-positive patients in stages I–III CRC ranges from 6–8% to 29%, raising questions about whether these differences are solely attributable to variations in disease stage. In addition, such technical variability in assays could further impact results due to the fact that clinical trials from different countries or research groups preferentially use one test over another, introducing an additional layer of uncertainty.

On a positive note, technological advancements continue to create opportunities for innovation. The potential use of aggregate biomarkers rather than individual mutations is gaining traction. For instance, epigenetic profiles and fragmentomics are already being implemented, while other aggregate biomarkers, such as mutational signatures and copy number profiles [67,94], show promise in differentiating CRC patients from healthy donors [58,59,61]. These emerging approaches benefit from increasingly affordable and accurate sequencing technologies, paving the way for more comprehensive and standardized MRD detection in the future.

## 5. Conclusions

The liquid biopsy revolution appears to be within reach in the context of MRD. The multitude of ongoing and recently completed clinical trials focused on this area underscores the significant interest of the scientific community in leveraging liquid biopsy to stratify patients into MRD+ and MRD− groups. This progress reflects the potential of MRD detection to transform post-surgical management by guiding personalized treatment decisions, reducing overtreatment, and intensifying therapy where it is most needed.

However, the high level of interest is tempered by the substantial technical, design, and performance variability among assays available for liquid biopsy-based MRD detection. The heterogeneity of these assays, encompassing differences in methodology, target selection, and sensitivity, poses significant challenges. This heterogeneity risks producing inconsistent results that may not solely reflect biological or clinical differences but could also stem from the preferential use of certain tests by specific countries or research groups, introducing an additional layer of uncertainty. Standardization and harmonization of MRD assays will be crucial to ensure that this promising technology delivers reliable and clinically actionable outcomes for patients. The path to the liquid biopsy revolution requires navigating these complexities, but the potential benefits for personalized oncology are undeniable.

## Figures and Tables

**Figure 1 genes-16-00071-f001:**
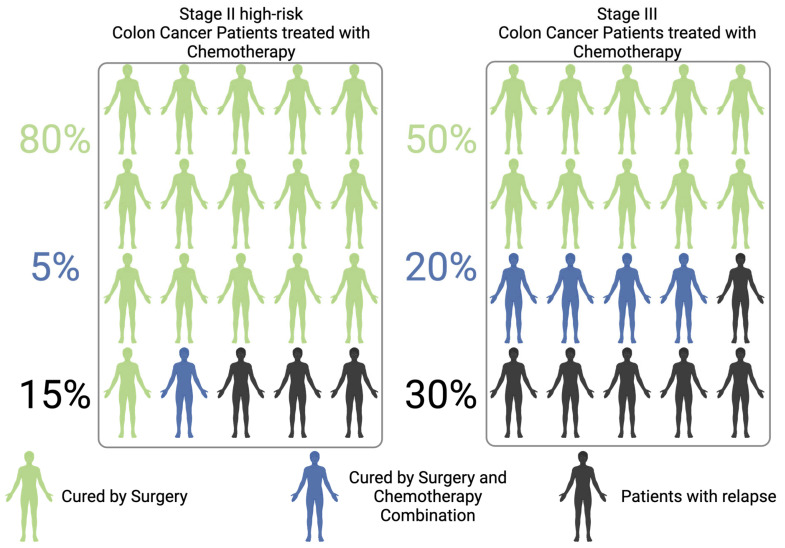
Scheme of Colon Cancer patients who are cured by surgery alone, or surgery/chemotherapy combination, or who relapse after resection.

**Figure 2 genes-16-00071-f002:**
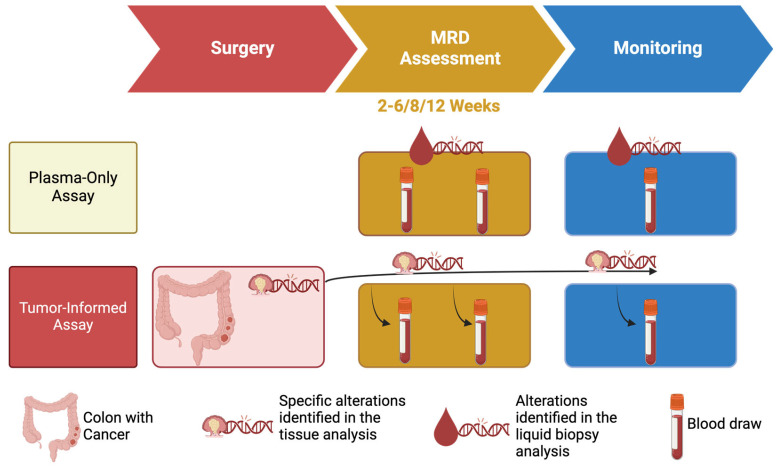
Schematic differences of plasma-only and tumor-informed assays.

**Table 1 genes-16-00071-t001:** Features and limitations of products for minimal residual disease identification using liquid biopsy.

Type of Assay	Assay	Company	Target Genomic Region in the Tissue	Number of Variants Monitored in the Blood	Features
Plasma-Only	Guardant Reveal	Guardant Health, Redwood, CA, USA	Not applicable	limited genomic target (previous version) and epigenomic regions, only 20 K epigenomic regions (2024 version)	It primarily utilizes advanced techniques, including methylation patterns and fragmentomics, to achieve high sensitivity without prior tumor tissue sequencing.
Plasma-Only	AVENIO ctDNA Surveillance Kit V2	Roche, Basel, Basel-Stadt, Switzerland	Not applicable	It analyses 197 genes, covering 471 frequently mutated regions associated with diseases, including genes listed in the National Comprehensive Cancer Network (NCCN) guidelines.	The kit employs iDES error suppression for high-accuracy detection of four mutation classes (SNV, indel, CNA, and fusion) in plasma-only samples, covering frequently mutated regions and supporting MRD monitoring.
Plasma-Only	PredicineALERT™ MRD assay	Predicine, Hayward, CA, USA	Not applicable	Targeted panel for minimal residual disease assessment	Blood and urine samples are supported.
Limitations of Plasma-Only		(A) metastases located in specific organs are known to release low levels of ctDNA. (B) smaller genetic targets may fail to capture the specific genetic alterations of each type of tumor.			
Tumor-informed	Signatera	Natera, Austin, TX, USA	Whole-Exome Sequencing (WES)	16 (clonal somatic variants)	The personalized tumor signature is monitored in plasma for high sensitivity and specificity.
Tumor-informed	Plasma Detect	Labcorp, Burlington, NC, USA	Whole-Genome Sequencing (WGS)	Robust patient-specific mutation set	It leverages WGS and machine learning for a tumor-informed MRD detection approach.
Tumor-informed	Haystack MRD	Haystack Oncology, Baltimore, MD, USA	Tumor-specific targeted panel	Custom (patient-specific mutations)	It uses bespoke sequencing panels to detect residual, recurrent, or resistant disease in plasma.
Tumor-informed	RaDaR™	NeoGenomics, Fort Myers, FL, USA	Tumor-specific targeted panel	Up to 48 tumor-specific variants	It focuses on MRD detection in solid tumors with precise monitoring using NGS technology.
Tumor-informed	MRDetect	Landau lab, New York, NY, USA	Whole-Genome Sequencing (WGS)	Whole-Genome Sequencing (WGS)	-
Limitations of Tumor-informed		(A) metastases located in specific organs are known to release low levels of ctDNA. (B) clonal mutations in the sequenced tissue could not accurately reflect the entire tumor’s genetic makeup. (C) there are disappearances of specific mutations that were originally used to define the tumor signature. (D) tumor tissue could be unavailable for the analysis.			

**Table 2 genes-16-00071-t002:** MRD positivity and the type of assay that was performed in clinical trials. Matched analysis = same assay was performed in blood/tissue; NA: not available; w: weeks.

PMID	First Author (Year)	Stage Study	Tumor-Informed/ Plasma-Only	Commercial/ Custom Assay	Weeks After Surgery	MRD+	Country
34001194	Chen et al. (2021) [81]	II–III	Matched Analysis	Custom	1 w	Stage II–III: 8.3%	China
33926918	Parkih et al. (2021) [85]	I–IV	Plasma-Only	Guardant Health	4 w	Stage I–IV: 14.56%	USA
34083233	Taieb et al. (2021) [82]	III	Plasma-Only	Custom	Before ACT	Stage III: 13.8%	France
34731746	Benhaim et al. (2021) [83]	II–III	Plasma-Only	Custom	1 w	Stage II–III: 10.5%	France
NA	Lonardi et al. (2023) [72]	II–III	Plasma-Only	Guardant Health	2–4 w	Stage II–III: 26%	Italy & Spain
NA	Morris et al. (2024) [90]	II	Plasma-Only	Guardant Health	2–12 w	Stage II: 5.54%	USA
27384348	Tie et al. (2016) [89]	II	Tumor-Informed	Custom	4–10 w	Stage II: 8.7%	Australia
28600478	Scholer et al. (2017) [80]	I–IV	Tumor-Informed	Custom	1–4 w	Stage I–III: 28.57%	Denmark
31070691	Reinert et al. (2019) [86]	I–III	Tumor-Informed	Signatera	4 w	Stage I–III: 10.6%	Denmark
31562764	Tarazona et al. (2019) [48]	I–III	Tumor-Informed	Custom	6–8 w	Stage I–III: 20.3%	Spain
31621801	Tie et al. (2019) [88]	III	Tumor-Informed	Custom	4–10 w	Stage III: 21%	Australia
34625408	Henriksen et al. (2022) [87]	III	Tumor-Informed	Signatera	2–4 w	Stage III: 14.28%	Denmark-Spain
35636041	Li et al. (2022) [84]	III	Tumor-Informed	Roche Avenio	2–4 w	Stage III: 15.9%	China
35657320	Tie et al. (2022) [16]	II	Tumor-Informed	Custom	4–7 w	Stage II: 15.46%	Australia
36646802	Kotani et al. (2023) [77]	I–IV	Tumor-Informed	Signatera	4 w	Stage I–IV: 18%	Japan
NA	Rubio-Alarcon et al. (2023) [91]	III	Tumor-Informed	PlasmaDetect	0–6 w	Stage III: 17.1%	Netherlands
NA	Dasari et al. (2023) [92]	II–IV	Tumor-Informed	Signatera	0–12 w	Stage II–IV: 15%	USA
NA	Kasi et al. (2024) [93]	I–IV	Tumor-Informed	Signatera	2–4 w	Stage II–III: 15.6%	USA

## Data Availability

The original contributions presented in this study are included in the article. Further inquiries can be directed to the corresponding author.

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
