# Peer review of "Liquid Biopsy and Challenge of Assay Heterogeneity for Minimal Residual Disease Assessment in Colon Cancer Treatment"

_genes, 2025, doi:10.3390/genes16010071_

Round 1
Reviewer 1 Report
Comments and Suggestions for Authors
The review is well written. The references are appropriately cited.
Comments:
1. Is any difference among countries which perform assays and clinical trials?
2. Figure 1: from which source or study? #5 reference? If it is from #5 article, what is the difference between this Figure 1 and the data used in #5 article?
3. Table 1: please add "limitation of each assay".
4. Table 2: please add the country which perform clinical trials.
5. Table 2: Please add "stage of clinical trial".
6. Please list all abbreviations.
7. What would be the future direction within 5 years and the major limitation of current review?
Author Response
The review is well written. The references are appropriately cited.
I thank the reviewer for their positive feedback and for recognizing the effort I put into the manuscript.
Comments:
- Is any difference among countries which perform assays and clinical trials?
I'm grateful for this smart question. A thorough review of the clinical trials reveals that there is a preference for certain approaches based on the geographic location of the studies. For example, French studies exhibit a tendency to favour custom plasma-only approaches, as illustrated in the works by Taieb et al. (2021) and Benhaim et al. (2021). Conversely, Danish studies employed tumor-informed approaches, as evidenced by Scholer et al. (2017), Reinert et al. (2019), and Henriksen et al. (2022), with Signatera being the assay of choice in two out of three cases. Furthermore, the three studies conducted by Tie and colleagues (2016, 2019, 2022) also favour a tumor-informed approach, with their use of the Safe-SeqS assay being noteworthy. This assay serves exploratory purposes to identify mutations to be tracked in plasma through a custom assay (see updated version of the Table 2).
Despite the limited number of trials included in the analysis (typically two or three per country), the findings suggest that there are research group preferences for specific approaches and assays that may be influenced by their individual experiences and areas of expertise. The Australian case, exemplified by the consistent use of custom tumor-informed assays in Tie's studies, is particularly emblematic of this trend.
I have incorporated this result into the updated manuscript, where it is presented in the Results section and further discussed in the Discussion and Conclusions sections. Additionally, as this is a relevant finding, I have slightly revised the abstract to include this finding for better visibility and coherence throughout the manuscript. Thank you again for this useful suggestion.
- Figure 1: from which source or study? #5 reference? If it is from #5 article, what is the difference between this Figure 1 and the data used in #5 article?
I would like to express my gratitude for the opportunity to elucidate the Figure 1. As the referee accurately observed, the work by Taieb et al. (Cancer Treat Rev 2019) served as a source of inspiration for this figure. However, it is crucial to note the distinctions between the two figures. The original Figure 1 from Taieb et al. presents data exclusively for stage III colon cancer. In contrast, the present figure extends and complements this information by also including data for high-risk stage II colon cancer. Additionally, my figure emphasizes the impact of cancer stage on the frequencies. Specifically, the data showed that patients who are cured by surgery alone account for around 80% of stage II patients and around 50% of stage III patients. In contrast, patients who are cured by a combination of surgery and chemotherapy account for around 5% of stage II patients and around 20% of stage III patients. Finally, the data showed that around 15% of stage II patients and around 30% of stage III patients experienced relapse after resection.
To ensure comprehensive citation, the following references have been added to the Figure 1 in the updated version of the manuscript: O'Connell JB et al., J Natl Cancer Inst 2004; Collienne M and Arnold D, Cancers 2020; Taieb J et al., Cancer Treat Rev 2019; André T et al. N Engl J Med. 2004; Quasar Collaborative Group et al. Lancet. 2007. These references provide the comprehensive background for the data presented in my figure, clarifying its originality and scope beyond the data reported by Taieb et al 2019.
- Table 1: please add "limitation of each assay".
The limitation of each assay has been incorporated into the new version of Table 1, as indicated by the referee.
- Table 2: please add the country which perform clinical trials.
I would like to express my gratitude for the commentary provided. In response, I incorporated a column that delineates the "country" of the study as delineated by the referee.
- Table 2: Please add "stage of clinical trial".
In Table 2, I have incorporated a column that delineates the "stage of clinical trial," as delineated by the referee. Furthermore, I have maintained the indication of the stage in the "MRD+" column. This is due to the fact that there are instances where MRD positivity is publicly reported for specific stages, rather than for all the stages encompassed by the study (Kasi et al. 2024 and Scholer et al. 2017).
- Please list all abbreviations.
As recommended by the referee, the list of abbreviations has been appended to the end of the new version of the revised manuscript.
- What would be the future direction within 5 years and the major limitation of current review?
The primary limitation of this review lies in the inability to access detailed technical specifications of individual commercial MRD products. Due to intellectual property protection, companies often do not disclose these details. As a result, while I aimed to provide a precise evaluation of the differences between various commercial products, it is not always possible to explore the specific technical details of each product. Consequently, the present study mainly focuses on the two overarching approaches underlying these assays: tumor-informed and plasma-only. Although it is uncertain what new technologies might be incorporated into commercial assays in the future, I anticipate that methods leveraging copy number alterations, fragmentomics, and mutational signature analysis will be instrumental in further lowering detection limits and overcoming some of the current limitations of MRD assays.
In the context of MRD, the "liquid biopsy revolution" is within reach in the context of MRD (within 5 years). This optimism is based on the multitude of ongoing and recently completed clinical trials in this area, which highlight the significant interest of the scientific community in using liquid biopsy to stratify patients into MRD+ and MRD- groups. This progress underscores the transformative potential of MRD detection to revolutionize post-surgical management by guiding personalized treatment decisions, reducing overtreatment, and intensifying therapy for patients who need it most.
The limitation described above has been added to the Discussion section in the updated manuscript.

Reviewer 2 Report
Comments and Suggestions for Authors
This is a really interesting and thoughtful review paper on liquid biopsy challenges regarding the MRD in CC. I believe minor revisions are required before the manuscript can proceed further. Thus, please answer or consider the following:
(1) Just to make sure, is the authorship list complete? Judging by the “Acknowledgments” section, it is a case. If yes, then congratulations to the Author for such hard work!
(2) Introduction, line 51: add “Colon Cancer” before “Patients” in the Figure 1 title.
(3) Section 1 or 2: I would like to see some simple comparison of liquid and traditional biopsy. Think about procedure invasiveness, ability to reflect the tissue architecture and cellular characteristics, information about tumor proteome, risks and costs, as well as reproducibility issues. A few additional sentences would be enough.
(4) Section 2, line 137: Figure 2 needs a more pronounced presentation, either in the figure’s description or in the figure itself (in that case, provide annotations for specific graphical elements such as colon, DNA helices, blood drop, vials etc.).
(5) Section 2, line 139: add a hyphen between “tumor” and “agnostic”. Please double-check the entire paper to make sure that “tumor-agnostic” or “tumor-informed” are with a hyphen.
(6) Section 2, line 158: please italicize “in silico” and make sure that similar examples (e.g., “in vitro”, “in vivo”, etc., if present) are also italicized.
(7) Section 2, line 164: why genomic target regions are not reported known for PredicineALERT™ MRD? Are they secured by law or something similar?
(8) Table 1: in the fourth column (“Target Genomic Region in the tissue”), consider changing the hyphen to italicized “unknown” or “not reported” or similar.
(9) Section 2, line 194: explain “ddPCR” on first use (digital droplet PCR). Do the same for regimens such as CAPOX, FOLFOX, etc.
(10) Remove empty rows in lines 243 and 391.
(11) Table 2: consider changing “na” to italicized “NA”.
(12) Conclusions, line 401: add a full stop.
Author Response
This is a really interesting and thoughtful review paper on liquid biopsy challenges regarding the MRD in CC. I believe minor revisions are required before the manuscript can proceed further.
I would like to express my sincere gratitude to the reviewer for considering my review thoughtful and interesting, as well as for providing valuable suggestions to enhance the quality of my work. I have addressed the comments and suggestions to improve the manuscript quality.
Thus, please answer or consider the following:
(1) Just to make sure, is the authorship list complete? Judging by the “Acknowledgments” section, it is a case. If yes, then congratulations to the Author for such hard work!
I would like to express my gratitude for the kind congratulations, which fill me with pride. It is always rewarding when one’s work is recognized. Concerning the authorship list, I can confirm that it is complete and that I am the sole author. I included the “Acknowledgments” section because I wanted to express my gratitude to the exceptional individuals I coordinate at the computational level and share the office with.
These colleagues have been with me since 2022 and continually inspire me through our scientific discussions. Their diverse scientific backgrounds and unique personalities enrich our collaborations, and their presence has been invaluable to my work. I felt it was important to highlight their contributions and express my heartfelt thanks for their ongoing support and the stimulating environment they help create.
Once again, I truly appreciate your thoughtful comment.
(2) Introduction, line 51: add “Colon Cancer” before “Patients” in the Figure 1 title.
I have done, thank you.
(3) Section 1 or 2: I would like to see some simple comparison of liquid and traditional biopsy. Think about procedure invasiveness, ability to reflect the tissue architecture and cellular characteristics, information about tumor proteome, risks and costs, as well as reproducibility issues. A few additional sentences would be enough.
Thank you for this constructive feedback. I agree that a comparison between liquid and traditional biopsies can increase the value to the manuscript by providing a broader perspective on the strengths and limitations of the liquid biopsy approach. I have appended several sentences to the initial part of the section 2 which was previously dedicated to the description of liquid biopsy.
The sentences in question are as follows:
“Liquid biopsy is a minimally invasive method that extracts genetic and multi-omic information from a simple blood draw [24, 25]. It is cost-effective method of medical analysis that is less invasive than traditional tissue analysis methods having the capacity to overcome the limitations of tissue-based methodologies, such as the effects of tumor heterogeneity and sampling bias [26, 27]. However, like all technologies, liquid biopsy has its limitations. It cannot provide information about tissue architecture and some cellular characteristics, and it may suffer from variability in ctDNA shedding [25, 28-30].”
(4) Section 2, line 137: Figure 2 needs a more pronounced presentation, either in the figure’s description or in the figure itself (in that case, provide annotations for specific graphical elements such as colon, DNA helices, blood drop, vials etc.).
I added the legend for specific graphical elements in the Figure 2.
(5) Section 2, line 139: add a hyphen between “tumor” and “agnostic”. Please double-check the entire paper to make sure that “tumor-agnostic” or “tumor-informed” are with a hyphen.
I have reviewed the entire manuscript and ensured that "tumor-agnostic" and "tumor-informed" are consistently written with a hyphen throughout the text. The necessary modifications have been made accordingly.
(6) Section 2, line 158: please italicize “in silico” and make sure that similar examples (e.g., “in vitro”, “in vivo”, etc., if present) are also italicized.
I checked the entire manuscript, modifying the word as requested by the reviewer.
(7) Section 2, line 164: why genomic target regions are not reported known for PredicineALERT™ MRD? Are they secured by law or something similar?
Thank you for raising this important question. In the context of MRD assays, oncologists require a product that can discriminate between a micrometastatic patient (MRD+) and one without detectable tumor information (MRD-) in the plasma. To achieve this, most commercial MRD products, including plasma-only assays like PredicineALERT™ MRD, do not disclose detailed information about the genomic targets or tumor-specific biomarkers detected. These details are often proprietary, as they are integral to the product's design and functionality. Such information is considered commercially sensitive and is often protected by intellectual property rights to safeguard the uniqueness of the assay and its competitive advantage in the market. This is particularly true for plasma-only commercial products, where the precise methodology and target regions are critical to the assay's performance and market differentiation.
Nonetheless, the study by Chen et al. (ref #33 in the updated version of the manuscript) described the platform PredicineATLAS™ (2022 version) as being based on NGS technology with a hybrid capture panel covering of approximately 600 genes.
I have included this additional information in the updated manuscript.
(8) Table 1: in the fourth column (“Target Genomic Region in the tissue”), consider changing the hyphen to italicized “unknown” or “not reported” or similar.
Thank you for this helpful suggestion. I have replaced the hyphen (“-”) with “not applicable” for Plasma-Only products, as these are usually not designed for use with tissue samples. The updated terminology ensures clarity and better aligns with the intended use of these products. The revised table has been incorporated into the manuscript.
(9) Section 2, line 194: explain “ddPCR” on first use (digital droplet PCR). Do the same for regimens such as CAPOX, FOLFOX, etc.
The acronym was initially reported in its extended form on the first use, and the Abbreviation list section was subsequently added at the end of the updated version of the manuscript.
(10) Remove empty rows in lines 243 and 391.
I deleted the blank lines.
(11) Table 2: consider changing “na” to italicized “NA”.
I modified the table 1 as recommended.
(12) Conclusions, line 401: add a full stop.
I added the full stop at the end of the Conclusion, thank you.
Round 2
Reviewer 1 Report
Comments and Suggestions for Authors
No more comments